# L-Tryptophan Enhances Intestinal Integrity in Diquat-Challenged Piglets Associated with Improvement of Redox Status and Mitochondrial Function

**DOI:** 10.3390/ani9050266

**Published:** 2019-05-22

**Authors:** Jingbo Liu, Yong Zhang, Yan Li, Honglin Yan, Hongfu Zhang

**Affiliations:** 1School of Life Science and Engineering, Southwest University of Science and Technology, Mianyang 621010, China; liuswust@163.com; 2State Key Laboratory of Animal Nutrition, Institute of Animal Sciences, Chinese Academy of Agricultural Sciences, Beijing 100000, China; zyzlrzjh@swust.edu.cn (Y.Z.); sq9517@126.com (Y.L.); zhanghfcaas@gmail.com (H.Z.)

**Keywords:** tryptophan, oxidative stress, intestinal barrier function, mitochondria, piglets

## Abstract

**Simple Summary:**

In the present study, three groups of piglets were treated with diquat, a bipyridyl herbicide which can utilize molecular oxygen to generate superoxide anion radicals and is widely considered as an effective chemical agent for inducing oxidative stress. The three groups were fed a 0, 0.15%, and 0.30% tryptophan (Trp) supplemented diet, and one control group without diquat treatment was used to study the protective effects of supplemented Trp on growth performance and intestinal barrier function of piglets exposed to oxidative stress. The results showed that 0.15% Trp supplementation alleviated diquat-induced impaired growth performance, intestinal barrier injury, redox imbalance, and mitochondrial dysfunction. These findings from the current study suggest that piglets under the condition of stress might need more Trp to maintain intestinal integrity and optimal growth performance, but the proper dosage of Trp supplementation is needed to determine for different conditions or models.

**Abstract:**

Tryptophan (Trp) supplementation has been shown to improve growth performance and enhance intestinal integrity in piglets. However, the effects of dietary Trp supplementation on the intestinal barrier function in piglets exposed to oxidative stress remain unknown. This study was conducted to evaluate whether dietary Trp supplementation can attenuate intestinal injury, oxidative stress, and mitochondrial dysfunction of piglets caused by diquat injection. Thirty-two piglets at 25 days of age were randomly allocated to four groups: (1) the non-challenged control; (2) diquat-challenged control; (3) 0.15% Trp-supplemented diet + diquat; (4) 0.30% Trp supplemented diet + diquat. On day seven, the piglets were injected intraperitoneally with sterilized saline or diquat (10 mg/kg body weight). The experiment lasted 21 days. Dietary supplementation with 0.15% Trp improved growth performance of diquat-challenged piglets from day 7 to 21. Diquat induced an increased intestinal permeability, impaired antioxidant capacity, and mitochondrial dysfunction. Although dietary supplementation with 0.15% Trp ameliorated these negative effects induced by diquat challenge that showed decreasing permeability of 4 kDa fluorescein isothiocyanate dextran, increasing antioxidant indexes, and enhancing mitochondrial biogenesis. Results indicated that dietary supplementation with 0.15% Trp enhanced intestinal integrity, restored the redox status, and improved the mitochondrial function of piglets challenged with diquat.

## 1. Introduction

The early-weaning strategy has been generally applied in pig production, because it can shorten the slaughter cycle of pigs and improve the reproduction performance of sows [1]. However, weaning per se is a stressful condition, which can cause an increased occurrence of diarrhea, thereby increasing mortality and reducing growth performance in piglets [2]. Moreover, weaning stress-induced intestinal barrier dysfunction, characterized by increased intestinal permeability, has been shown to be one of the primary causes of severe diarrhea and reduced growth performance of post-weaning piglets [3]. Tryptophan (Trp) is the second-limiting amino acid in most corn-based diets of pigs, and functionally regulates body protein synthesis and improves stress and immune responses [4,5]. Several studies have shown that dietary supplementation with Trp enhances intestinal integrity and reduces diarrhea rate, thereby improving growth performance [6,7]. In contrast, a previous study showed that dietary supplementation with 0.1% Trp significantly increased intestinal permeability and decreased the gene expression of tight junction proteins [8]. Interestingly, the authors from the same group, which concluded that a Trp-supplemented diet impaired the intestinal integrity of piglets, contended that a 0.15% Trp-supplemented diet increased villus height to crypt depth ratio without affecting intestinal permeability [9]. A more recent study stated that weaning piglets fed the 0.20% or 0.4% Trp-supplemented diet exhibited enhanced intestinal mucosal barrier function than those fed the basal diet [5]. Therefore, it can be concluded that a proper dosage of Trp supplemented in the diet would be favorable for the enhancement of intestinal integrity and the improvement of growth performance of nursery piglets.

Intestinal redox imbalance has been reported to contribute to weaning-induced impaired intestinal mucosal barrier and reduced growth performance in piglets [10]. Weaning stress has been shown to disrupt the free-radical metabolism and compromise the antioxidant system, thereby inducing severe oxidative stress, which is characterized by insufficient antioxidants and antioxidant enzymes in clearing an excessive generation of reactive oxygen species (ROS) [11,12]. Several studies have shown that oxidative stress and the disruption of redox homeostasis lead to an increase in intestinal paracellular permeability by rupture of the intestinal barrier and/or by decline of the expression of tight junction proteins [13,14,15]. To preserve cellular integrity and function, the intestine needs a high amount energy, which is mainly provided by the intracellular organelles, mitochondria [16]. Meanwhile, the mitochondria are the major source of intracellular ROS as well as one of the important targets for the damaging effect of ROS [17]. Under the situation of oxidative stress, the antioxidant capacity of cells was not sufficient to clear the produced ROS, resulting in an excess accumulation of ROS in mitochondria [13]. The imbalance between ROS production and its clearance contributed to mitochondrial dysfunction, like abnormal biogenesis process and disruption of ATP synthesis [18]. In vivo studies demonstrated that the maintenance of intestinal barrier function is dependent on mitochondrial biogenesis and ATP production [13,19]. Taken together, the compromised intestinal redox homeostasis and mitochondrial dysfunction may contribute to the impaired intestinal barrier function in piglets exposed to oxidative stress.

Considering the effects that the addition of Trp had on intestinal barrier function of weaning piglets, determining whether dietary supplementation with proper dosage of Trp can exert such an effect in an oxidative stress-challenged model is imperative. In addition, it would be of great interest to determine whether the benefits of Trp supplementation on intestinal barrier function is associated with improvement in the intestinal redox status and mitochondrial function. In a previous study, dietary supplementation of 0.12% Trp enhanced antioxidant capacity and increased antioxidant enzymes’ activities in the liver of oxidative stress-challenged piglets [12]. Therefore, we hypothesized that dietary supplementation with proper dosage of Trp could alleviate oxidative stress-induced impaired intestinal barrier function through restoring redox homeostasis and mitochondrial function. In the present study, a well-documented model for inducing oxidative stress by injecting diquat [12,13] was adopted to determine the protective effects of Trp supplementation on intestinal barrier function, redox status, and mitochondrial function in diquat-challenged piglets.

## 2. Materials and Methods

### 2.1. Ethical Statement

This experiment was conducted in accordance with the Chinese guidelines for animal welfare, and all experimental procedures were approved by the Ethics Committee of the Southwest University of Science and Technology (Mianyang, Sichuan, China) under permit number DKX-1020150040.

### 2.2. Animals and Treatments

A total of 32 castrated male piglets, which are Duroc × (Landrace × Yorkshire) commercial hybrid pigs, from 8 litters (4 piglets per litter) were selected from a high-sanitary-status breeder farm of New Hope Group [20]. Piglets were weaned at 18 days of age and brought to the animal experimental facilities at the Swine Research Unit of the Southwest University of Science and Technology. After 7 days of adaptation, selected piglets (25 days of age) with an average body weight (BW) of 6.62 kg were randomly allocated to one of the 4 treatment groups (*n* = 8 piglets per treatment): (1) the non-challenged control group (control), piglets fed a basal diet and injected with saline; (2) diquat-challenged group (diquat), piglets fed the basal diet and injected with diquat (Sigma-Aldrich, St Louis, MO, USA); (3) 0.15% Trp + diquat, piglets fed the basal diet supplemented with 0.15% Trp (Trp1 diet) and injected with diquat; (4) 0.30% Trp + diquat, piglets fed the basal diet supplemented with 0.30% Trp (Trp2 diet) and injected with diquat. Treatments were balanced for weight and litter and were set to explore the protective effects of Trp on performance and intestinal integrity of piglets under the condition of oxidative stress, and thus the groups, within which piglets were fed the Trp-supplemented diets without diquat challenge, were not included in the present study. The basal diet, of which the standard ileal digestible (SID) Trp content was 0.22, was formulated to meet or exceed National Research Council-recommended nutrient requirements for pigs weighing 7–11 kg (NRC 2012) [21]. The basal diet was supplemented with 0.153% or 0.306% of L-tryptophan (98.0% purity; Evonik Degussa (China) Co., LTD, Beijing, China) to make the Trp1 or Trp2 diet containing 0.15% or 0.30% more SID Trp than the basal diet. The supplemental levels of Trp were based on a previous study showing that 0.12% supplemental Trp could alleviate diquat-induced oxidative stress of the liver [12]. All the diets were kept isonitrogenous and isoenergetic by adjusting L-alanine and cornstarch contents, as previously described [22]. The ingredient composition and nutrient level of experimental diets are listed in Table 1. The piglets were fed with their respective diets for a 21-d period. On day 7, the piglets in diquat, 0.15% Trp + diquat, and 0.30% Trp + diquat groups were injected intraperitoneally with diquat (dibromide monohydrate, Sigma-Aldrich, St. Louis, MO, USA) at 10 mg/kg BW while the piglets in the control group received the same volume of sterilized saline. During the whole trial, the piglets were housed individually in stainless-steel metabolic cages equipped with a nipple drinker and a feeder and were allowed ad libitum access to feed and water. The piglets had no access to antibiotics throughout the experiment.

### 2.3. Performance Measurement and Sample Collection

The feed intake of each piglet was recorded daily to calculate average daily feed intake (ADFI) for period d0–d7 and period d7–d21, and the BW of each piglet was measured at d0, d7, and d21 of the trial to determine the average daily weight gain (ADG) for period d0–d7 and period d7–d21. Feed to gain ratio of each piglet during each period was determined by dividing the corresponding ADFI by ADG. In this study, the addition of L-tryptophan in the basal diet was to elucidate the benefits of tryptophan supplementation on intestinal dysfunction in diquat-challenged piglets. Thus, the group with the better growth performance (0.15% Trp + diquat) between the two tryptophan-treated groups was chosen as the proper tryptophan treatment to further study the benefits of tryptophan on intestinal function. On day 21 of the trial, 8 piglets each from the control group, diquat group, and 0.15% Trp + diquat group were sampled. Blood samples of the piglets were collected from the precaval vein into tubes without anticoagulant (5 mL) after overnight feed deprivation, followed by centrifugation at 3000× g for 10 min. The supernatant (serum) was collected and stored at −20 °C until further assays. After blood sampling, piglets were anaesthetized by an intravenous injection (10 mg/kg BW) of a mixture of zolazepam and tiletamine (a lyophilized product containing 125 mg tiletamine and 125 mg zolazepam, was dissolved in 5 mL of sterilized saline to a final concentration of 50 mg/mL just before use), followed by exsanguination. Immediately post-mortem, the entire small intestine was removed and the jejunum was obtained for sample collection. A 10-cm segment from the proximal jejunum was collected for Ussing chamber studies. Another 5-cm portion of the adjacent jejunum was opened longitudinally and cleaned with ice-cold phosphate buffer saline (PBS, pH 7.4). Approximately 100 mg of fresh jejunum sample from this segment was collected and stored at 4 °C for isolation of intestinal mitochondria. Furthermore, a 20-cm segment of the jejunum from the same region was processed to harvest mucosal scrapings, which were immediately immersed in liquid nitrogen and then stored at −80 °C for further analyses.

### 2.4. Serum Intestinal Permeability Biomarker Concentrations

Diamine oxidase (DAO) activity in serum was measured using spectrophotometry as previously described [23]. In brief, the reaction mixture, which contained 0.5 mL of serum sample, 3 mL of phosphate buffer (0.2 M, pH 7.2), 0.1 mL of o-dianisidine-methanol solution (0.5% of o-dianisidine in methanol), 0.1 mL of horseradish peroxidase solution (0.004%), and 0.1 mL of substrate solution (0.175% of cadaverine dihydrochloride), was incubated for 30 min at 37 °C, and the absorbance at 436 nm was measured to calculate DAO activity. The results were expressed as units per liter (U/L) of serum. The D-lactate concentration in serum was measured by adopting commercial ELISA kits (Beijing Luyuan Byrd biological technology Co., Ltd, Beijing, China) with a microplate reader, according to the manufacturer’s instructions. The result of D-lactate concentration was presented as microgram per milliliter (μg/mL). All assays were performed in triplicate.

### 2.5. Ex Vivo Measurement of Intestinal Barrier Function

Intestinal mucosal permeability was assessed ex vivo by measuring transepithelial electrical resistance (TER) and the translocation of macromolecular markers using the Ussing chamber technique as previously described [13,24,25]. Fresh segments of jejunum mucosa samples were stripped from the seromuscular layers and then mounted in the EasyMount Ussing chamber system (model VCC MC6, Physiological Instruments, San Diego, CA, USA). The intestinal sheets were surrounded by 5 mL of Ringer’s solution with 10 mM glucose on the serosal side and the same concentration of mannitol on the mucosal side. The system was oxygenated with 95% O_2_ and 5% CO_2_ airflow and water-jacketed to 37 °C. The clamps connecting to the Acquire and Analyze software (Physiologic Instruments, San Diego, CA, USA) were used for automatic data collection. After an equilibration period of 20 min, TER was recorded at a 20-min interval over 80 min. For measuring the translocation of macromolecular markers, after an equilibration period of 20 min, the 4 kDa fluorescein isothiocyanate dextran (FD4, Sigma-Aldrich, St. Louis, MO, USA) was added to the mucosal side of the mounted intestinal sheets to a final concentration of 0.8 mg/mL of FD4. Samples (50 μL) were taken from the serosal side of the mounted intestinal sheets at 20, 40, 60, and 80 min after adding the marker and then transferred into a 96-well assay plate. The FD4 fluorescence intensity of the samples, which were collected from the serosal side of tissue, were measured at an excitation wavelength of 485 nm using the Fluorescence Microplate Reader (FLx800, Bio-Tek Instruments Inc., Winooski, VT, USA). The FD4 flux over the 80-min period was calculated and presented as an apparent permeation coefficient (cm/s).

### 2.6. Intestinal Redox Status Measurements

Jejunal mucosa was used to measure superoxide dismutase (SOD), glutathione peroxidase (GPx), catalase (CAT) activities, and malondialdehyde (MDA) concentration by adopting commercial SOD, GPx, CAT, and MDA assay kits following the manufacturer’s instructions (Nanjing Jiancheng Bioengineering Institute, Jiangsu, China). Total protein content in jejunal mucosa was determined with a Pierce BCA Protein Assay Kit (Pierce Chemical, Rockford, IL, USA) according to the manufacturer’s protocol. The results of SOD, GPx, and CAT activities were expressed as units per microgram of jejunal protein (U/mg prot), while the result of MDA was presented as nmol per microgram of jejunal protein (nmol/mg prot). All assays were performed in triplicate.

### 2.7. Isolation of Intestinal Mitochondria

The isolation of jejunal mitochondria was performed by using a commercial Tissue Mitochondrial Isolation Kit (Beyotime Institute of Biotechnology, Shanghai, China) following the manufacturer’s instructions. All procedures for the isolation of mitochondria were conducted at 4 °C. The purity of isolated mitochondria was verified by checking the existence of β-actin and voltage-dependent anion-selective channel (VDAC) with Western blotting as previously described [26]. The β-actin and VDAC are marker proteins for cytosol and mitochondria, respectively. In the present study, no expression of β-actin in isolated mitochondria was detected, indicating no contamination of cytosol-derived protein in isolated mitochondria samples. The mitochondrial protein content was measured with the Pierce BCA Protein Assay Kit (Pierce Chemical, Rockford, IL, USA) according to the manufacturer’s protocol.

### 2.8. Intestinal Mitochondrial ROS Assay

The ROS production in intestinal mitochondria was measured using a fluorescence microplate reader with dichlorohydro-fluorescein diacetate (DCFDA) as described in a previous study [8]. Briefly, isolated intestinal mitochondria (0.4 mg/mL) were stained with 2 μM DCFDA and incubated at 25 °C for 20 min. DCFDA passed through the membrane where it was oxidized in the presence of ROS to dichlorohydro-fluorescein (DCF), of which the excitation wavelength and the emission wavelength were 485 nm and 530 nm, respectively. Fluorescence was determined at λex 485 nm and λem 530 nm. Therefore, the arbitrary units of fluorescence intensity of DCF was used to indicate the ROS level in mitochondrial samples. The ROS level of all samples was expressed as fold change relative to the average ROS level of the control group.

### 2.9. Intestinal Mitochondrial Membrane Potential (ΔΨm) Assay

The mitochondrial membrane potential (ΔΨm) from all isolated mitochondria samples were measured using the mitochondrial membrane potential assay kit with fluorescent dye JC-1 (Beyotime Institute of Biotechnology, Shanghai, China). The interaction of JC-1 dye with mitochondrial membrane components can alter its fluorescence properties associated with the switch in monomer and aggregate. The JC-1 monomer fluorescence form, which produces a green fluorescence, is seen when the mitochondrial membrane potential is low, whereas at high mitochondrial membrane potential, the aggregate form of JC-1 dye, which produces a red fluorescence, occurs. The isolated intestinal mitochondria (0.4 mg/mL) were stained with JC-1 (500 nM) at 37 °C for 30 min. The fluorescence intensity was determined with a fluorescence microplate reader. The JC-1 monomer fluorescence form was excited at 485 nm and the emission fluorescence was detected at 530 nm, the JC-1 aggregate form was excited at 485 nm, and the emission was detected at 590 nm. Consequently, ΔΨm was expressed by the red/green fluorescence intensity ratio. The intestinal mitochondrial membrane potential (ΔΨm) of all samples was expressed as fold change relative to the average ΔΨm of the control group.

### 2.10. Determination of Jejunal Mitochondrial DNA (mtDNA) Content

Total DNA was extracted from jejunal samples using a DNAiso Reagent (Takara Bio Inc., Dalian, China). The quality control of DNA was performed as previously described [27]. The content of mtDNA relative to nuclear genomic DNA was measured by co-amplifying the mitochondrial DNA-loop (*mt D-loop*) and the nuclear-encoded bet β-actin (*ACTB*) using real-time PCR assay. The mt D-loop and ACTB, of which the primers and probes sequences are listed in Table 2, were quantified by the fluorescent probes. Each PCR reaction consisted of 1 μL DNA template, 8 μL TaqMan Universal Master mix, 1 μl forward primer (2 μM), 1 μL reverse primer (2 μM), 1 μL enhance solution, and 7 μL Milli-Q water in a total volume of 20 μL. The thermal cycling conditions were as follows: an initial denaturation and enzyme activation step (95 °C for 10 s), then fifty cycles of denaturation/annealing/extension and data acquisition (95 °C for 5 s, 60 °C for 25 s, 72 °C for 10 s). The ratio of mtDNA to genomic DNA content was calculated as ΔCt (mt Cq _Dloop_ − nuclear Cq _ACTB_). The relative abundance (RE) indicated the factorial difference in mtDNA content between groups. The RE was calculated as 2^−ΔΔCq^ method, where ΔΔCt = ΔCq _mtDNA content in other group_ – ΔCq _mtDNA content in the control group_.

### 2.11. RNA Isolation and Reverse Transcription Quantitative PCR (RT-qPCR)

Total RNA from jejunal mucosa was extracted by using Bio-Rad Aurum Total RNA Fatty and Fibrous Tissue Kit (Bio-Rad Laboratories, Hercules, CA, USA). RNA quality control was performed before the synthesis of cDNA. The purity and concentration of isolated RNA samples were measured with the Nanodrop ND-1000 (Nanodrop Technologies, Thermo Scientific, Wilmington, DE, USA). The integrity of RNA was checked by agarose gel electrophoresis. The absence of genomic DNA contamination of RNA samples was verified by a minus-reverse transcription control PCR. After the quality control of RNA samples, 1 μg of DNA-free high-quality RNA was converted to cDNA by using the ImProm-II cDNA synthesis kit (Promega, Madison, WI, USA). The targeted genes, including tight junction proteins (zona occludens 1 (*ZO1*), zona occludens 2 (*ZO2*), occludin (*OCLN*), claudin 1 (*CLDN1*), and claudin 2 (*CLDN2*)), redox-sensitive genes (copper/zinc superoxide dismutase (*SOD1*), heme oxygenase 1 (*HMOX1*), glutathione peroxidase (*GPX1*), thioredoxin reductase (*TXNRD1*), nuclear respiratory factor 2 (*NRF2*), and kelch like ECH associated protein 1 (*KEAP1*)), and genes related to mitochondrial biogenesis (nuclear respiratory factor 1 (*NRF1*), transcription factor A, mitochondrial (*TFAM*), peroxisome proliferative activated receptor gamma coactivator 1 alpha (*PPARGC1A*), sirtuin 1 (*SIRT1*), single stranded DNA binding protein 1 (*SSBP1*), and DNA-directed RNA polymerase, mitochondrial (*POLRMT*)), were determined in the present study. The primers were designed with Primer3Plus based on the certain exon-exon boundaries of published gene sequences of pigs on the National Center for Biotechnology Information (NCBI) database and were listed in Table 2. The PCR reaction was comprised of 5 μL 2× KAPA SYBR FAST qPCR Kit Master Mix, 0.5 μL forward primer (5 μM), 0.5 μL reverse primer (5 μM), 2 μL Milli-Q water, and 2 μL cDNA template. The following thermal cycling conditions were used for RT-qPCR: an initial denaturation and enzyme activation step (95 °C for 3 min), then forty cycles of denaturation/annealing and data acquisition (95 °C for 20 s, 40 s at annealing temperature depending on primer), and melt curve analysis (from 70 °C to 90 °C with 0.5 °C increments every 5 s). Five points of 4-fold serial dilutions of cDNA were included in each run to obtain the PCR efficiency by constructing a standard curve. In the present study, PCR amplification efficiencies consistently ranged from 90% to 110% and were used to convert the Cq values into raw data. TATA-binding protein (*TBP*), DNA topoisomerase II beta (*TOP2B*), and *ACTB*, of which the primer sequences are listed in Table 3, were used as reference genes to normalize the raw data of RT-qPCR. The relative gene expression level was expressed as fold change relative to the average mRNA level of the gene in the control group.

### 2.12. Gel Electrophoresis and Western Blotting

The protein expression of ZO1, occludin, claudin1, HO1, and GPx in jejunal mucosa were further detected by Western blotting, performed as previously described [28,29], to confirm whether the transcriptional regulation also occurred at the translation level. Briefly, the total protein extractions from jejunal mucosa were harvested in lysis buffer (Beyotime Biotechnology, Jiangsu, China) supplemented with a complete protease inhibitor cocktail (Pierce Chemical, Rockford, IL, USA) and cleared by centrifugation at 12,000× g for 30 min at 4 °C. The protein content of each lysate was measured with a BCA Protein Assay Kit (Pierce Chemical, Rockford, IL, USA) on a plate reader. Protein lysates were separated (30 μg per lane) on 10% SDS-PAGE gel, and then proteins were transferred onto an polyvinylidine difluoride (PVDF) membrane (Millipore, Billerica, MA, USA). The membrane was washed in Tris-buffered saline containing tween (TBST) and blocked in 5% skimmed-milk solution (Beyotime Biotechnology, Jiangsu, China) at room temperature for one hour with gentle shaking. Afterwards, the blots were probed overnight at 4 °C with the respective primary antibody. Subsequently, the blots were washed with TBST and incubated with an horseradish peroxidase (HRP) conjugated secondary antibody for 1 h at room temperature. The intensity of the bands on the blots was quantified by the Image Lab statistical software (Bio-Rad Laboratories, Hercules, CA, USA). The relative expression of targeted protein was normalized using GAPDH as the internal protein and was presented as the fold change relative to the control group.

### 2.13. Statistical Analysis

A piglet was considered as the experimental unit for all analyses (*n* = 8 per treatment), and all data were expressed as mean ± standard error. The data of growth performance were performed with an analysis of variance (ANOVA) for repeated measures and others were subjected to ANOVA using SAS 9.1 software (SAS Institute, Cary, NC, USA). The differences between treatments were evaluated using Tukey’s test. Statistical significance was set at probability values <0.05.

## 3. Results

### 3.1. Growth Performance

Before the diquat challenge, during the first seven days of the trial, L-tryptophan supplementation had no effect on body weight, average daily weight gain, and feed to gain ratio (*p* > 0.05), whereas dietary supplementation with 0.15% Trp significantly increased average daily feed intake of piglets (*p* < 0.05). Intraperitoneal injection with diquat significantly decreased body weight on day 21, average daily weight gain from d7 to d21, average daily feed intake from d7 to d21, and feed to gain ratio from d7 to d21 for piglets fed the basal diet, as compared to non-challenged piglets (*p* < 0.05). There were no differences in the average daily weight gain, average daily feed intake, and feed to gain ratio during d7 to d21 between the control group and the 0.15% Trp + diquat group (*p* > 0.05). In addition, for diquat-challenged piglets, dietary supplementation with 0.15% Trp significantly increased body weight on day 21, average daily weight gain from d7 to d21, and average daily feed intake from d7 to d21 (*p* < 0.05). However, the average daily weight gain and feed to gain ratio from d7 to d21 of diquat-treated piglets were not influenced by dietary supplementation with 0.30% Trp (*p* > 0.05) (Table 4). The performance results suggest that dietary supplementation with 0.15% Trp protected piglets from diquat challenge-induced decline of growth performance.

### 3.2. Intestinal Permeability

In the present study, the intestinal permeability was evaluated by jointly determining the serum biomarkers, D-lactate concentration and DAO activity, and the TER and translocation of FD4 in the jejunum. There was no significant effect of diquat challenge, nor the L-tryptophan addition on serum D-lactate concentration (*p* > 0.05). Compared to the non-challenged piglets, diquat-treated piglets fed the basal diet exhibited higher serum DAO activity (*p* < 0.05). There was no significant difference in serum DAO activity between the control group and the 0.15% Trp + diquat group (*p* > 0.05). The ex vivo Ussing chamber study showed that diquat-challenged piglets fed the basal diet exhibited a lower TER and a higher FD4 flux from the mucosa side to the serosa side in the jejunum compared to those in the control group (*p* < 0.05). In diquat-treated piglets, dietary supplementation with 0.15% Trp significantly increased TER and decreased the flux of FD4 from the mucosa side to the serosa side in the jejunum (*p* < 0.05). In addition, diquat-treated piglets fed the 0.15% Trp supplemented diet had similar TER and FD4 flux to the non-challenged piglets (*p* > 0.05) (Figure 1).

### 3.3. mRNA Abundances and Protein Expression of Tight Junction Proteins in the Jejunum

The mRNA expression of tight junction proteins (*ZO1*, *ZO2*, *OCLN*, *CLDN1*, and *CLDN2*) in the jejunum can be found in Figure 2. Compared with the non-challenged piglets, diquat-treated piglets fed the basal diet had lower mRNA abundances of *ZO1*, *OCLN*, and *CLDN1* in jejunal mucosa (*p* < 0.05), whereas diquat-challenged piglets fed the diet supplemented with 0.15% Trp had similar levels of mRNA expression of tight junction proteins (*p* > 0.05). Dietary supplementation with 0.15% Trp significantly upregulated gene expression of *ZO1*, *OCLN*, and *CLDN1* in the jejunum of diquat-treated piglets (*p* < 0.05). To check whether the effects that dietary Trp supplementation had on mRNA levels of tight junction proteins also occurred at the translation level, the protein expression of ZO1, occludin, and claudin1 were measured, and the related results can be found in Figure 3. Exposure of piglets fed the basal diet to diquat significantly decreased protein expression of ZO1 and occludin in the jejunum, as compared to the non-challenged piglets (*p* < 0.05). There were no significant differences in protein levels of ZO1 and occludin in the jejunum between the non-challenged piglets and the diquat-treated piglets fed the 0.15% Trp-supplemented diet (*p* > 0.05). In diquat-challenged piglets, dietary supplementation with 0.15% Trp significantly upregulated protein expression of ZO1 and occludin in the jejunum (*p* < 0.05). No significant differences for other parameters were found.

### 3.4. Antioxidant Enzymes Activities and MDA Content in the Jejunum

Injection of diquat significantly decreased SOD, GPx, and CAT activities in jejunal mucosa of piglets fed the basal diets compared to non-challenged piglets (*p* < 0.05). Additionally, diquat-treated piglets fed the basal diet had a higher concentration of MDA in jejunal mucosa than piglets in the control group (*p* < 0.05). However, there were no differences in SOD and CAT activities and MDA in jejunal mucosa between piglets in the 0.15% Trp + diquat group and those in the control group (*p* > 0.05). The GPx activity in jejunal mucosa of piglets in the 0.15% Trp + diquat group was significantly lower than that of piglets in the control group (*p* < 0.05). In diquat-treated piglets, those fed the diet supplemented with 0.15% Trp exhibited increased SOD, GPx, and CAT activities and lower MDA concentration in jejunal mucosa than those fed the basal diet (*p* < 0.05) (Figure 4).

### 3.5. Redox-Sensitive Genes Expression in the Jejunum

The gene expression of *SOD1*, *HMOX1*, *GPX1*, *TXNRD1*, *NRF2*, and *KEAP1* is summarized in Figure 5. Diquat injection decreased mRNA abundances of *SOD1*, *HMOX1*, *GPX1*, *TXNRD1*, and NRF2 in jejunal mucosa of piglets fed the basal diet compared to non-challenged piglets (*p* < 0.05). However, there were no differences in gene expression of *SOD1*, *HMOX1*, *GPX1*, *TXNRD1*, and NRF2 in jejunal mucosa between piglets in the 0.15% Trp + diquat group and those in the control group (*p* > 0.05). In addition, piglets in the 0.15% Trp + diquat group exhibited higher mRNA levels of *SOD1*, *HMOX1*, and *GPX1* in jejunal mucosa than diquat-treated piglets fed the basal diet (*p* < 0.05). The protein expression of HO1 and GPx in jejunal mucosa can be found in Figure 6. Compared with the non-challenged piglets, diquat challenge decreased protein expression of HO1 and GPx in jejunal mucosa of piglets fed the basal diet (*p* < 0.05). In contrast, there was no difference in the protein level of HO1 and GPx in jejunal mucosa between the non-challenged piglets and the diquat-challenged piglets fed the 0.15% Trp-supplemented diet (*p* > 0.05). In diquat-treated piglets, Trp supplementation significantly upregulated protein expression of HO1 and GPx in jejunal mucosa (*p* < 0.05).

### 3.6. Intestinal Mitochondrial ROS Production and ΔΨm

In comparison to the non-challenged piglets, the mitochondrial ROS production was enhanced and the ΔΨm in the jejunum was declined in diquat-treated piglets fed the basal diet (*p* < 0.05). However, dietary supplementation with 0.15% Trp significantly decreased the mitochondrial ROS production and increased the ΔΨm in the jejunum of diquat-challenged piglets, as compared to the diquat group (*p* < 0.05). There were no differences in jejunal mitochondrial ROS production and ΔΨm between the control group and the 0.15% Trp + diquat group (*p* > 0.05) (Figure 7).

### 3.7. Expression of Genes Related to Mitochondrial Biogenesis in the Jejunum

The gene expression of *NRF1*, *TFAM*, *PPARGC1A*, *SIRT1*, *SSBP1*, and *POLRMT* is presented in Figure 8. The mRNA abundances of *NRF1*, *PPARGC1A*, and *SSBP1* in the jejunum were decreased in diquat-treated piglets fed the basal diet compared to the non-challenged piglets (*p* < 0.05). However, there were no differences in the gene expression of *NRF1*, *PPARGC1A*, and *SSBP1* in the jejunum between diquat-treated piglets fed the 0.15% Trp-supplemented diet and the non-challenged piglets (*p* > 0.05). Dietary supplementation with 0.15% Trp significantly upregulated the mRNA expression of *NRF1*, *PPARGC1A*, and *SSBP1* in the jejunum of diquat-treated piglets compared to piglets in the diquat group (*p* < 0.05). In addition, diquat-challenged piglets fed the 0.15% Trp-supplemented diet exhibited higher mRNA abundance of *TFAM* in the jejunum than those fed the basal diet (*p* < 0.05). There was no difference in the gene expression of *TFAM* in the jejunum between diquat-treated piglets fed either the basal diet or 0.15% Trp supplemented diet and the non-challenged piglets (*p* > 0.05).

### 3.8. mtDNA Content in the Jejunum

The mtDNA content in the jejunum was decreased in diquat-treated piglets fed the basal diet compared to saline-treated piglets fed the basal diet (*p* < 0.05). However, there was no difference in mtDNA content between diquat-treated piglets fed the 0.15% Trp-supplemented diet and non-challenged piglets (*p* > 0.05) (Figure 8).

## 4. Discussion

The importance of Trp to control feed intake in pigs is well-documented, the daily feed intake of weaning piglets was reported to be linearly increased as dietary tryptophan level ranging from 0.12% to 0.26% increased, which was associated with enhanced ghrelin secretion in the gastrointestinal tract [30]. A recent study showed that the basal diet, with a dietary Trp level of 0.21% supplemented with 0.2% or 0.4% Trp increased the average feed daily intake and average daily gain of weaning piglets, but a 0.2% Trp-supplemented diet exhibited stronger growth promotion than a 0.4% Trp-supplemented diet [31]. Consistently, in the present study, the addition of 0.15% Trp rather than 0.30% Trp to the basal diet, of which the Trp level is 0.22%, increased feed intake of weaning piglets before diquat challenge, which might be associated with increased ghrelin secretion and less aggressive behavior of piglets [30]. Of note, the National Research Council recommended that the tryptophan requirement for pigs weighing 7–11 kg is 0.22% [21]. It can be concluded that adding a proper dosage of Trp to the recommended level facilitates the ingestion of feed in weaning piglets under normal conditions. Diquat challenge has been shown to induce the decline of average daily gain and average daily feed intake [12,13], and dietary supplementation with 0.12% Trp prevented the oxidative stress-induced reduction in growth performance [12]. In the present study, dietary supplementation with 0.15% Trp, rather than 0.30% Trp, protected piglets from diquat challenge-induced reduction in growth performance, suggesting the proper dosage of supplemented Trp is 0.15% for diquat-treated piglets. A previous study showed that the optimal ratio of dietary Trp to Lys ratio was 0.26 for weaning piglets [32]. In the present study, the ratio of dietary Trp to Lys in a 0.15% Trp-supplemented diet was 0.27, which was similar to a meta-analysis indicating that the requirements of Trp to Lys ratio of pigs from 7 to 30 kg were 0.22, 0.22, and 0.20 with average daily gain, average daily feed intake, and feed to gain ratio as response criteria, respectively [33]. In addition, high dietary Trp to Lys ratio or diquat challenge has been shown to decrease the nitrogen utilization as well as reduce the intestinal absorption and whole-body metabolism of other amino acids, resulting in poor growth performance of weaning piglets [12,34]. The results in this study showed that a 0.15% Trp-supplemented diet rather than a 0.30% Trp-supplemented diet improved the growth performance of piglets exposed to oxidative stress, which might be associated with the more proper dietary Trp to Lys ratio in Trp1 (0.15% Trp supplementation) diet.

The integrated intestinal barrier plays a critical role in preventing pathogens and harmful materials from entering the body through the intestine, and thus is important for the host’s health [8,9]. In the present study, diquat-treated piglets exhibited higher serum DAO activity than non-challenged piglets. DAO, an intracellular enzyme mainly synthesized in the intestinal epithelia of animals, is released into the blood when the intestinal barrier is injured [35]. Therefore, blood DAO activity was used as a circulating biomarker to evaluate the intestinal barrier function [36]. In addition, the intestinal barrier function was also measured by the Ussing chamber technique to detect the TER and paracellular flux of FD4. TER and FD4 are primarily used to study permeation through the paracellular pathway and can reflect the opening of tight junctions [37,38]. Under stress conditions, weaning stress, or oxidative stress, the intestinal barrier function of piglets was impaired, as evidenced by decreased TER and increased flux of FD4 [10,13]. In the current study, diquat treatment decreased jejunal TER and increased jejunal flux of FD4. The data from this study, both indirect and direct evidences, indicate that the administration of diquat impaired the intestinal barrier function of piglets. As a nutritionally essential amino acid, Trp has been shown to regulate the expression of tight junction proteins and promote intestinal protein synthesis to enhance the intestinal mucosal barrier in pigs [39]. In agreement with this study, dietary supplementation with 0.15% Trp protected piglets from diquat-induced impaired intestinal mucosa barrier function, as indicated by decreased serum DAO activity, increased jejunal TER, and decreased jejunal FD4 flux. The intestinal barrier is physically composed of epithelial cells connected by tight junction proteins, such as ZOs, claudin families, and occludin, regulating the selective permeability between epithelial cells [40]. The expression of tight junction proteins is important for maintaining the intestinal epithelial barrier [41]. Previous studies showed that oxidative stress downregulated the expression of tight junction proteins [13,37]. Consistently, in this study, diquat challenge decreased mRNA expression of *ZO1*, *OCLN*, and *CLDN1* as well as the protein expression of ZO1 and occludin. The family of ZO is a part of the cytoplasmic plaque of the tight junction proteins, and occludin is a unique marker of tight junction integrity found in the epithelial barrier; its presence or absence could reflect the permeability of the intestinal epithelium [24]. The higher jejunal tight junction permeability in diquat-treated piglets might be attributed to the lower expression of ZO1 and occludin, which was proven in a previous study showing that the lower expression of *OCLN* contributed to the increased intestinal permeability in piglets born with intrauterine growth restriction (IUGR) [42]. Consistent with the improvement of intestinal barrier function, a diet supplemented with 0.15% Trp reserved the decreased abundances of tight junction proteins in the jejunum induced by diquat treatment. An in vitro study involving intestinal porcine epithelial cells demonstrated that dietary Trp enhanced the expression of tight junction proteins [39]. In addition, dietary Trp supplementation has been shown to prevent the reduction in mRNA expression of tight junction proteins in the small intestine of piglets infected by *Escherichia coli* [43]. Thus, dietary tryptophan supplementation could alleviate the intestinal barrier damage induced by diquat-oxidative stress. Previous studies have demonstrated that microbiota-derived tryptophan metabolites, such as indole, which are ligands for the aryl hydrocarbon receptor (AhR), have a major role in intestinal homeostasis and have been implicated in the pathogenesis of metabolic syndrome and inflammatory bowel disease [31,44]. Dietary Trp supplementation has been shown to enhance the production of AhR ligands by enriching Trp-metabolizing bacteria in the small intestine of piglets, thereby beneficially affecting mucosal integrity [5,31]. Given that the abundances of Trp-metabolizing bacteria are reduced by oxidative stress, it can be speculated that dietary Trp supplementation might enhance the intestinal integrity of oxidative stress-challenged piglets by restoring the microbiota composition. However, this hypothesis should be confirmed in future research.

Several studies have shown that the regulatory contribution to the barrier tightness of tight junction proteins can be downregulated by oxidants and enhanced by antioxidants [45,46,47]. Dietary Trp supplementation has been demonstrated to increase the production of tryptophan metabolites and elevate the activity tryptophan-metabolizing enzymes, such as tryptophan 2,3-dioxygenase, which exhibited antioxidant activity [12]. To explore whether the beneficial effects of dietary Trp supplementation on intestinal barrier function of diquat-treated piglets is associated with the improved redox status, antioxidant enzymes activity, MDA content, and the expression of redox-sensitive genes in jejunal mucosa were determined. In the present study, diquat challenge decreased the activities of SOD and GPx and increased MDA concentration in the jejunum of piglets fed the basal diet, which was consistent with previous studies [12,13,37]. SOD, is the soluble Cu/Zn SOD localized in the intermembrane or cytosol, and has been shown to convert the superoxide into hydrogen peroxide (H_2_O_2_). GPx is a family of enzymes providing a protective mechanism against oxidative stress by scavenging ROS [48]. The decreased activities of SOD and GPX indicated that the excess of superoxide may happen in the jejunum of diquat-treated piglets. Additionally, MDA, a secondary product of lipid oxidation, has been considered as a maker to monitor lipid peroxidation [49]. The increased MDA content in the jejunum demonstrated that the severe lipid peroxidation may occur in diquat-treated piglets. In line with a previous study on IUGR piglets [42], the *SOD1*, *HMOX1*, *GPX1*, *TXRND1*, and *NRF2* expression showed a downregulation in diquat-treated piglets. *NRF2* has been reported to regulate the levels of endogenous antioxidant enzymes against oxidative stress. It has been shown that *HMOX1*, an important antioxidant enzyme regulated by *NRF2*, is crucial in mitigating oxidative stress in intestinal epithelial cells by regulating the ROS levels [49]. In the present study, the protein expression of HO1 is downregulated in the jejunum of diquat-challenged piglets, giving further proof for the compromised antioxidant capacity in the case of oxidative stress. *TXNRD1* is an enzyme keeping thioredoxin in a reduced state, during which the produced ROS are cleared [50]. The lower expression of TXNRD1 may also reflect a disturbed antioxidant capacity of diquat-challenged piglets. These results indicated that the diquat treatment induced the disruption of the antioxidant system and imbalanced the redox state in piglets. In the present study, dietary Trp supplementation increased activities of SOD, GPx, and CAT, mRNA expression of *SOD1*, *HMOX1*, and *GPX1*, as well as the protein expression of HO1, while it decreased the MDA content in the jejunum of diquat-treated piglets. In a previous study, Trp was found to increase antioxidant defense enzyme activities in serum and liver and to attenuate diquat-induced oxidative stress [12], thus corroborating with the results of the present study. Therefore, dietary supplementation with a proper dosage of Trp effectively ameliorated the oxidative stress in the jejunum of the piglets that were challenged with diquat. It has been shown that some tryptophan metabolites, such as 5-hydroxytryptophan and 3-hydroxyanthranilic acid, have an antioxidant capacity [51]. Furthermore, recent studies showed that gut microbiota might be involved in the beneficial effects that tryptophan has on the intestinal function of the host [5,31]. Hence, in future studies, it would be of great interest to determine the underlying mechanisms through which Trp can improve the redox status of piglets challenged with diquat, and to investigate whether Trp metabolites and gut microbiota are involved in this process.

It has been shown that mitochondria are sensitive to oxidative damage because they lack protective histones, and oxidative stress has been shown to induce mtDNA damage [52]. In the present study, diquat challenge reduced the mtDNA content in the jejunum of piglets, which is in line with the previous study [37]. Several studies have reported that mtDNA damage is associated with increased ROS generation by mitochondria and decreased ΔΨm of mitochondria. Our data showed that the ROS production in jejunal mitochondria was increased in piglets treated by diquat, which is consistent with previous studies showing that ROS level increased in cardiac or intestinal mitochondria under the condition of oxidative stress [13,18,53]. An in vitro study confirmed that diquat-induced disruption of the antioxidant enzyme system caused insufficient ROS degradation, resulting in an excess of ROS accumulation in intestinal epithelial cells [54]. In addition, the overproduction of ROS has been shown to trigger the opening of the inner membrane anion channel, resulting in mitochondrial membrane depolarization, which is characterized by decreased ΔΨm [18]. It is believed that the compromised redox homeostasis is associated with mitochondrial DNA damage and overproduction of mitochondrial ROS. In the present study, dietary Trp supplementation decreased the mitochondrial ROS production and increased mitochondrial ΔΨm, thereby restoring the mtDNA content in the jejunum of piglets challenged by diquat. According to the antioxidant property of Trp, the possible reason for the lower production of ROS in mitochondria may be that dietary Trp supplementation enhanced the clearance of mitochondrial ROS in piglets challenged by diquat. A previous study demonstrated that the alteration of mtDNA content is in accordance with the changes of expression of transcriptional factors involved in mitochondrial biogenesis [26]. Furthermore, it has been shown that the alteration of ROS production and ΔΨm of intestinal mitochondria is caused by an abnormal mitochondrial biogenesis process [13]. NRF1 and PPARGC1A, the major regulators of mitochondrial biogenesis, have been shown to be downregulated in the jejunum of piglets exposed to oxidative stress [13]. Another previous study showed that IUGR piglets, when exposed to severe oxidative stress, had a lower expression of *NRF1* and *PPARGC1A* in the jejunum [26]. In line with previous studies, our data showed that diquat challenge downregulated *NRF1* and *PPARGC1A* in the jejunum. In addition, *SSBP1*, important for mtDNA replication, was shown to downregulate in the jejunum of piglets treated with diquat [13,26]; our study revealed similar results. It has been shown that normal mitochondrial biogenesis and function are required for the maintenance of intestinal barrier function [19]. Therefore, it can be speculated that diquat-induced intestinal barrier injury may be attributed to mitochondrial dysfunction. We hypothesized that the protective effect of dietary tryptophan on an intestinal barrier injury induced by diquat was exerted by improving the mitochondrial function. In the present study, dietary supplementation with Trp prevented piglets from diquat-induced mtDNA damage, overproduction of ROS, and compromised mitochondrial biogenesis process. These findings implied that dietary Trp supplementation could enhance mitochondrial biogenesis to attenuate the intestinal barrier injury by oxidative stress.

Collectively, dietary supplementation with a proper dosage of Trp prevented oxidative stress-induced impaired growth performance and intestinal barrier function. The results from this study suggest that dietary Trp supplementation improved intestinal barrier function of diquat-treated piglets by enhancing the activities and mRNA or protein expression of antioxidant enzymes, and ameliorating the mtDNA damage as well as upregulating the transcriptional factors involved in mitochondrial biogenesis.

## 5. Conclusions

Diquat treatment induced increased intestinal permeability and compromised redox homeostasis of piglets. Dietary supplementation with 0.15% Trp improved growth performance, enhanced intestinal integrity, restored redox balance, and improved mitochondrial function of diquat-treated piglets.

## Figures and Tables

**Figure 1 animals-09-00266-f001:**
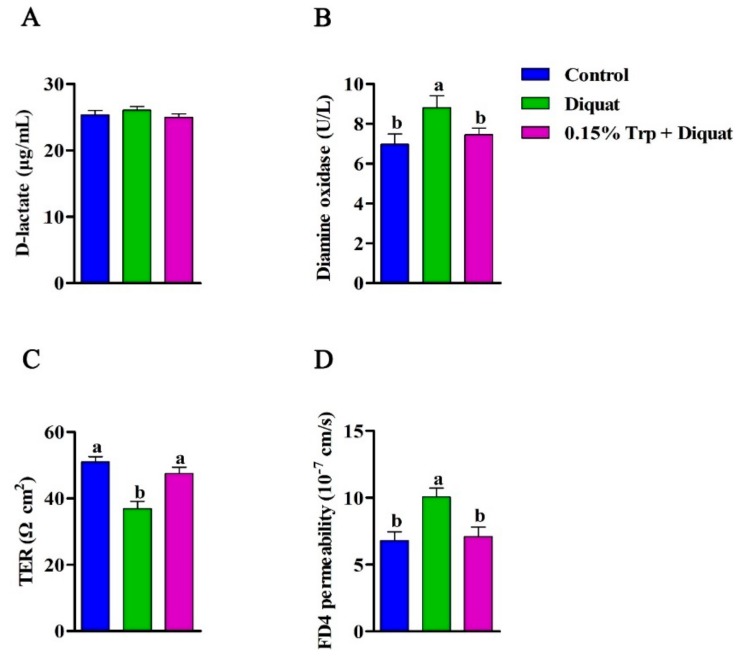
Effects of dietary tryptophan supplementation on intestinal permeability of piglets challenged with diquat. (**A**) D-lactate concentration; (**B**) diamine oxidase activity; (**C**) TER, transepithelial electrical resistance; (**D**) the flux of FD4, 4 kDa fluorescein isothiocyanate dextran, at the jejunum. Data are means ± standard error; *n* = 8 for each group. Mean values sharing different superscripts within the same row differ significantly (*p* < 0.05). Control, piglets fed the basal diet without diquat injection; diquat, piglets fed the basal diet and challenged with diquat; 0.15% Trp + diquat, piglets fed the 0.15% supplemented diet and treated with diquat.

**Figure 2 animals-09-00266-f002:**
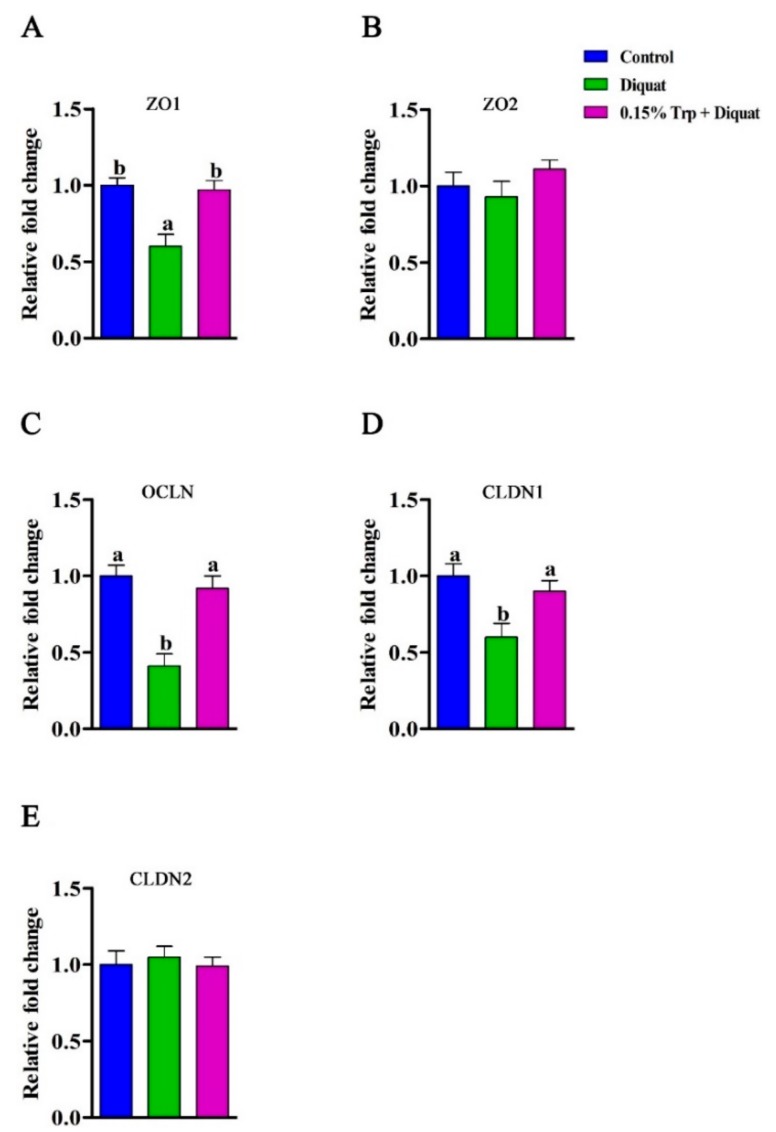
Effects of dietary tryptophan supplementation on mRNA expression of tight junction proteins in piglets challenged with diquat. (**A**) *ZO1*, zona occludens1; (**B**) *ZO2*, zona occludens 2; (**C**) *OCLN*, occludin; (**D**) *CLDN1*, claudin1; (**E**) *CLDN2*, claudin 2. Data are means ± standard error; *n* = 8 for each group. Mean values sharing different superscripts within the same row differ significantly (*p* < 0.05). Control, piglets fed the basal diet without diquat injection; diquat, piglets fed the basal diet and challenged with diquat; 0.15% Trp + diquat, piglets fed the 0.15% supplemented diet and treated with diquat.

**Figure 3 animals-09-00266-f003:**
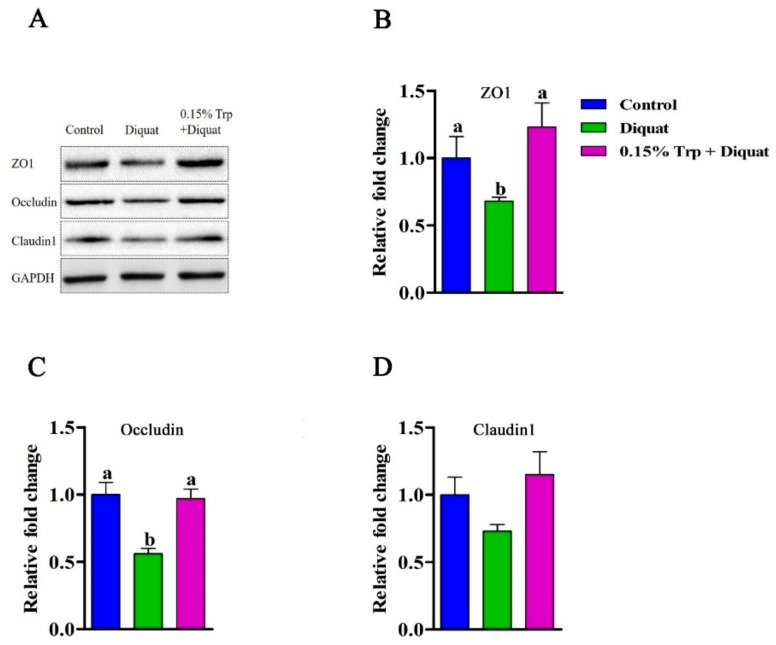
Effects of dietary tryptophan supplementation on protein expression of tight junction proteins in piglets challenged with diquat. (**A**) Representative blots of ZO1, occludin, claudin1, and GAPDH in the jejunal mucosa of piglets; (**B**) protein expression of ZO1, zona occludens1; (**C**) protein expression of occludin; (**D**) protein expression of claudin1. Data are means ± standard error; *n* = 8 for each group. Mean values sharing different superscripts within the same row differ significantly (*p* < 0.05). Control, piglets fed the basal diet without diquat injection; diquat, piglets fed the basal diet and challenged with diquat; 0.15% Trp + diquat, piglets fed the 0.15% supplemented diet and treated with diquat.

**Figure 4 animals-09-00266-f004:**
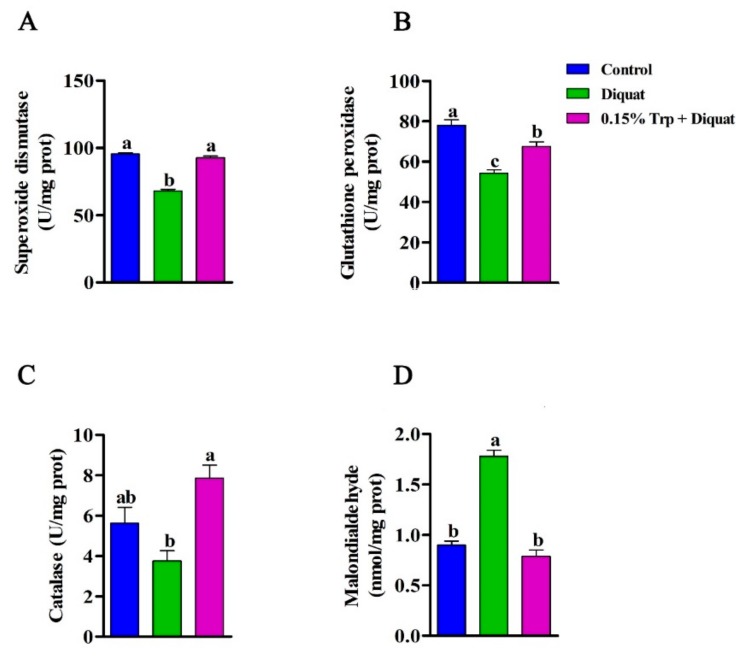
Effects of dietary tryptophan supplementation on jejunal redox status of piglets challenged with diquat. (**A**) Superoxide dismutase activity; (**B**) glutathione peroxidase activity; (**C**) catalase activity; (**D**) malondialdehyde concentration. Data are means ± standard error; *n* = 8 for each group. Mean values sharing different superscripts within the same row differ significantly (*p* < 0.05). Control, piglets fed the basal diet without diquat injection; diquat, piglets fed the basal diet and challenged with diquat; 0.15% Trp + diquat, piglets fed the 0.15% supplemented diet and treated with diquat.

**Figure 5 animals-09-00266-f005:**
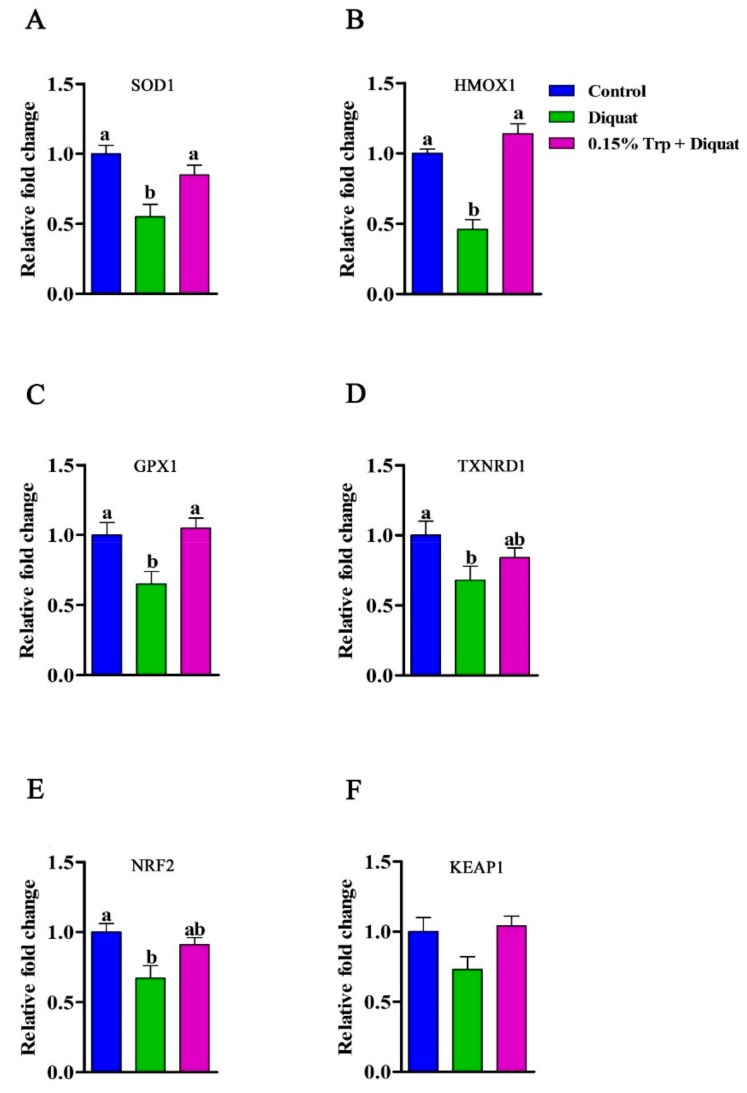
Effects of dietary tryptophan supplementation on mRNA expression of redox sensitive genes in piglets challenged with diquat. (**A**) *SOD1*, copper/zinc superoxide dismutase; (**B**) *HMOX1*, heme oxygenase 1; (**C**) *GPX1*, glutathione peroxidase; (**D**) *TXNRD1*, thioredoxin reductase; (**E**) *NRF2*, nuclear respiratory factor 2; (**F**) *KEAP1*, kelch like ECH associated protein 1. Data are means ± standard error; *n* = 8 for each group. Mean values sharing different superscripts within the same row differ significantly (*p* < 0.05). Control, piglets fed the basal diet without diquat injection; diquat, piglets fed the basal diet and challenged with diquat; 0.15% Trp + diquat, piglets fed the 0.15% supplemented diet and treated with diquat.

**Figure 6 animals-09-00266-f006:**
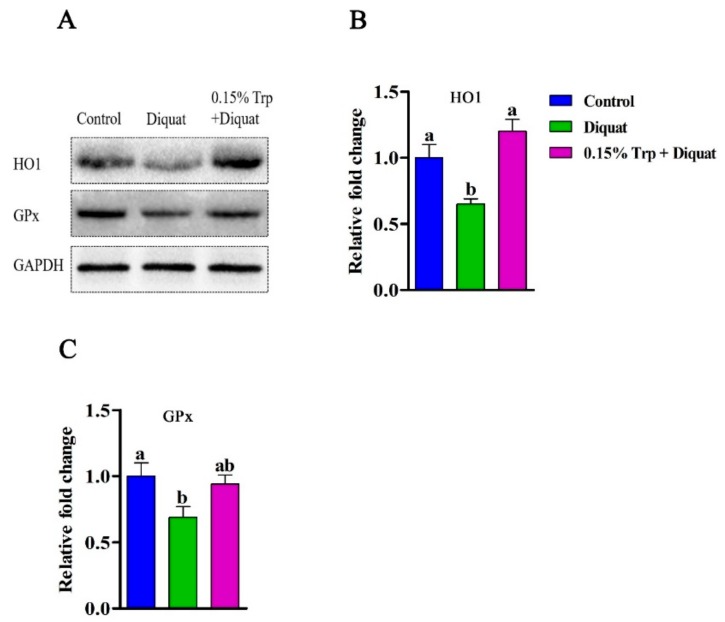
Effects of dietary tryptophan supplementation on protein expression of antioxidant enzymes in piglets challenged with diquat. (**A**) Representative blots of HO1, GPx, and GAPDH in the jejunal mucosa of piglets; (**B**) HO1, heme oxygenase 1; (**C**) GPx, glutathione peroxidase. Data are means ± standard error; *n* = 8 for each group. Mean values sharing different superscripts within the same row differ significantly (*p* < 0.05). Control, piglets fed the basal diet without diquat injection; diquat, piglets fed the basal diet and challenged with diquat; 0.15% Trp + diquat, piglets fed the 0.15% supplemented diet and treated with diquat.

**Figure 7 animals-09-00266-f007:**
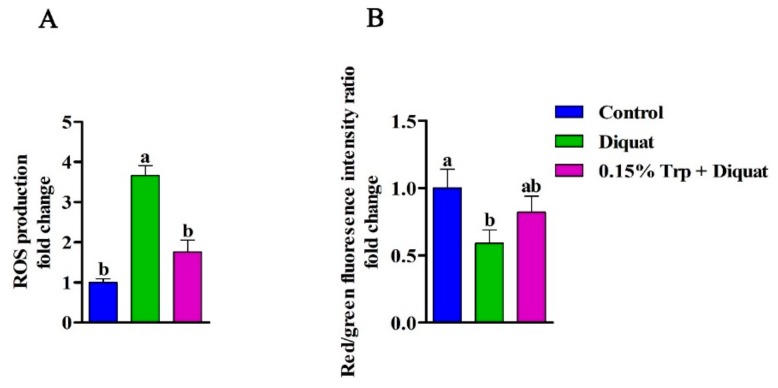
Effects of dietary tryptophan supplementation on intestinal mitochondrial reactive oxygen species (ROS) production (**A**) and intestinal mitochondrial membrane potential change (**B**) of piglets challenged with diquat. The ROS production and mitochondrial membrane potential were expressed as fold changes, calculated relative to the control group. Data are means ± standard error; *n* = 8 for each group. Mean values sharing different superscripts within the same row differ significantly (*p* < 0.05). Control, piglets fed the basal diet without diquat injection; diquat, piglets fed the basal diet and challenged with diquat; 0.15% Trp + diquat, piglets fed the 0.15% supplemented diet and treated with diquat.

**Figure 8 animals-09-00266-f008:**
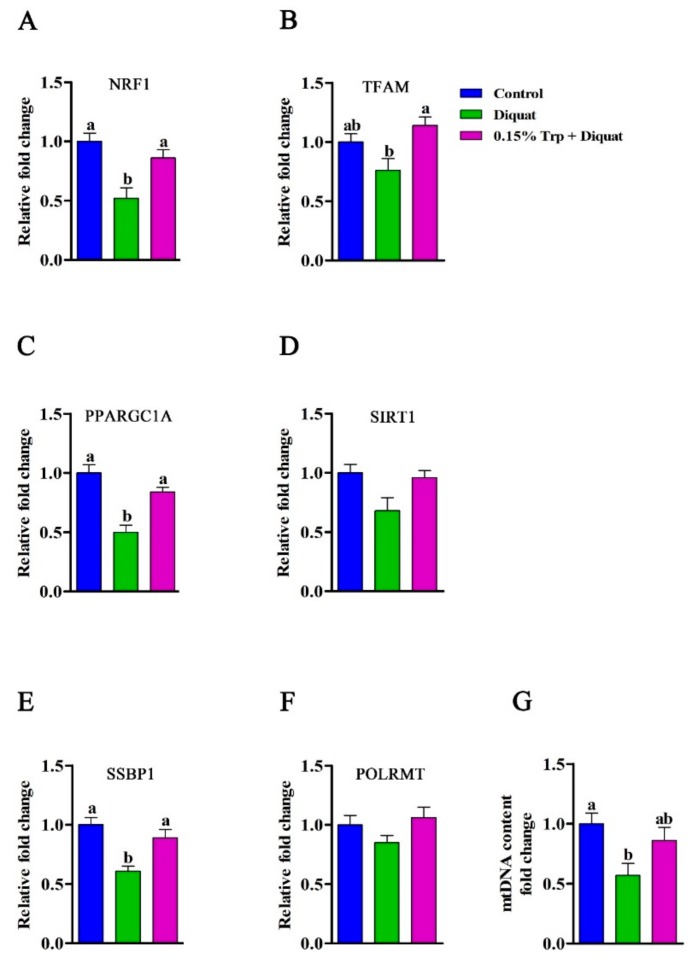
Effects of dietary tryptophan supplementation on mRNA expression of genes involved in mitochondrial biogenesis and mtDNA content in the jejunum of piglets challenged with diquat. (**A**) *NRF1*, nuclear respiratory factor 1; (**B**) *TFAM*, transcription factor A, mitochondrial; (**C**) *PPARGC1A*, peroxisome proliferative activated receptor gamma coactivator 1 alpha; (**D**) *SIRT1*, sirtuin 1; (**E**) *SSBP1*, single stranded DNA binding protein 1; (**F**) *POLRMT*, DNA-directed RNA polymerase, mitochondrial; (**G**) mtDNA, mitochondrial DNA. Data are means ± standard error; *n* = 8 for each group. Mean values sharing different superscripts within the same row differ significantly (*p* < 0.05). Control, piglets fed the basal diet without diquat injection; diquat, piglets fed the basal diet and challenged with diquat; 0.15% Trp + diquat, piglets fed the 0.15% supplemented diet and treated with diquat.

**Table 1 animals-09-00266-t001:** Ingredients and chemical composition of experimental diets.

Items	Basal Diet	Trp1 Diet ^3^	Trp2 Diet ^3^
Ingredients composition, %			
Corn	30.83	30.83	30.83
Extruded corn	24.00	24.00	24.00
Extruded soybean	6.00	6.00	6.00
Dehulled soybean meal (44% crude protein)	10.00	10.00	10.00
Soybean protein concentrate	10.00	10.00	10.00
Whey powder	5.00	5.00	5.00
Fish meal	5.00	5.00	5.00
Soybean oil	1.80	1.80	1.80
Sucrose	3.00	3.00	3.00
Glucose	1.69	1.69	1.69
Cornstarch	0.039	0.019	0
Salt	0.30	0.30	0.30
Limestone	0.72	0.72	0.72
CaHPO4	0.52	0.52	0.52
Chloride choline	0.10	0.10	0.10
Vitamin premix ^1^	0.05	0.05	0.05
Mineral premix ^2^	0.30	0.30	0.30
L-Trptophan (98.0%)	0	0.153	0.306
L-Alanine (99.8%)	0.271	0.138	0.004
DL-Methionine (98.5%)	0.13	0.13	0.13
L-Lysine·HCl (78%)	0.18	0.18	0.18
L-Threonine (98.5%)	0.07	0.07	0.07
Total	100.00	100.00	100.00
Nutrient level (Calculated value)			
DE/(MJ/kg)	15.04	15.04	15.04
Crude protein (%)	20.88	20.88	20.88
Ca (%)	0.80	0.80	0.80
Available phosphorous (%)	0.41	0.41	0.41
SID Lys (%)	1.35	1.35	1.35
SID Met (%)	0.46	0.46	0.46
SID Thr (%)	0.80	0.80	0.80
SID Trp (%)	0.22	0.37	0.52

^1^ The vitamin premix provided the following per kg of the diet: retinyl acetate, 0.95 mg; cholecalciferol, 0.02 mg; DL-α-tocopheryl acetate, 28 mg; menadione, 2 mg; riboflavin, 1mg; pyridoxine, 3 mg; vitamin B12, 48 μg; D-pantothenic acid, 30 mg; folic acid, 2.0 mg; biotin, 1mg. ^2^ The mineral premix provided the following per kg of the diet: Fe (ferrous sulfate), 100 mg; Cu (copper sulfate), 6 mg; Mn (manganese oxide), 4 mg; Zn (zinc sulfate), 100 mg; I (potassium iodate), 0.14 mg; Se (odium selenite), 0.35 mg. ^3^ Trp1 diet, basal diet supplemented with 0.15% Trp; Trp2 diet, basal diet supplemented with 0.30% Trp.

**Table 2 animals-09-00266-t002:** Primers and probes sequences used for determination of mitochondrial DNA (mtDNA) content.

Gene Symbols	Nucleotide Sequence (5′-3′)	Accession No.
*ACTB*	F: CCCCTCCTCTCTTGCCTCTC R: AAAAGTCCTAGGAAAATGGCAGAAG P: (FAM) TGCCACGCCCTTTCTCACTTGTTCT (Eclipse)	DQ452569
*mt D-loop*	F: GATCGTACATAGCACATATCATGTC R: GGTCCTGAAGTAAGAACCAGATG P: (FAM) CCAGTCAACATGCGTATCACCACCA (Eclipse)	AF276923

mtDNA, mitochondrial DNA; *ACTB*, β-actin; *mt D-loop*, mitochondrial DNA-loop.

**Table 3 animals-09-00266-t003:** Nucleotide sequences of primers used to measure targeted genes.

Gene Symbols	Nucleotide Sequence of Primers (5′–3′)	Accession No.
*ACTB*	F: TCTGGCACCACACCTTCT R: TGATCTGGGTCATCTTCTCAC	XM_003124280.3
*TOP2B*	F: AACTGGATGATGCTAATGATGCT R: TGGAAAAACTCCGTATCTGTCTC	NM_001258386.1
*TBP*	F: GATGGACGTTCGGTTTAGG R: AGCAGCACAGTACGAGCAA	DQ178129
*ZO1*	F: ATCTCGGAAAAGTGCCAGGA R: CCCCTCAGAAACCCATACCA	XM_003480423.3
*ZO2*	F: CCAGGAAGCACAGAATGCAA R: AAGTCTGGCGGGACCTCTCT	XM_005660148.2
*OCLN*	F: CATGGCTGCCTTCTGCTTCATTGC R: ACCATCACACCCAGGATAGCACTCA	NM_001163647.2
*CLDN1*	F: TATGACCCCATGACCCCAGT R: GCAGCAAAGTAGGGCACCTC	NM_001244539.1
*CLDN2*	F: TTCCTCCCTGTTCTCCCTGA R: CACTCTTGGCTTTGGGTGGT	NM_001161638.1
*SOD1*	F: GGTCCTCACTTCAATCCTGAATCC R: CACACCATCTTTGCCAGCAGT	NM_0011190422
*HMOX1*	F: CGCTCCCGAATGAACACTCT R: GCGAGGGTCTCTGGTCCTTA	NM_001004027.1
*GPX1*	F: TGCTCATTGAGAACGTAGCGT R CAGGATCTCCCCATTCTTCGC	NM_214201.1
*TXNRD*	F: GTGCTGAGGAGCTTCCCGAGATGT R: TCCAGGACCATGACCCGCTTGTTAA	NM_214154.3
*NRF2*	F: CCCATTGAGGGCTGTGATCT R: GCCTTCAGTGTGCTTCTGGTT	NM_031789.2
*KEAP1*	F: GGCTGGGATGCCTTGTAAAG R: GGGCCCATGGATTTCAGTT	NM_057152.2
*NRF1*	F: GCCAGTGAGATGAAGAGAAACG R: CTACAGCAGGGACCAAAGTTCAC	AK237171.1
*TFAM*	F: GGTCCATCACAGGTAAAGCTGAA R: ATAAGATCGTTTCGCCCAACTTC	AY923074.1
*PPARGC1A*	F: CCCGAAACAGTAGCAGAGACAAG R: CTGGGGTCAGAGGAAGAGATAAAG	NM 213963
*SIRT1*	F: TGACTGTGAAGCTGTACGAGGAG R: TGGCTCTATGAAACTGCTCTGG	EU030283.2
*SSBP1*	F: CTTTGAGGTAGTGCTGTGTCG R: CTCACCCCTGACGATGAAGAC	AK352341.1
*POLRMT*	F: CTTTGAGGTTTTCCAGCAGCAG R: GCTCCCAGTTTTGGTTGACAG	XM 001927064.1

ACTB, β-actin; TOP2B, DNA topoisomerase II beta; TBP, TATA-binding protein; ZO1/2, zona occludens 1/2; OCLN, occludin; CLDN1/2, claudin1/2; SOD1, copper/zinc superoxide dismutase; HMOX1, heme oxygenase 1; GPX1, glutathione peroxidase; TXNRD1, thioredoxin reductase; NRF2, nuclear respiratory factor 2; KEAP1, kelch like ECH associated protein 1; NRF1, nuclear respiratory factor 1; TFAM, transcription factor A, mitochondrial; PPARGC1A, peroxisome proliferative activated receptor gamma coactivator 1 alpha; SIRT1, sirtuin 1; SSBP1, single stranded DNA binding protein 1; POLRMT, DNA-directed RNA polymerase, mitochondrial.

**Table 4 animals-09-00266-t004:** Effects of dietary tryptophan supplementation on the growth performance of piglets challenged with diquat.

Items	Control	Diquat	0.15% Trp + Diquat	0.30% Trp + Diquat
Body weight (kg)				
Day 0	6.62 ± 0.20	6.63 ± 0.21	6.62 ± 0.21	6.63 ± 0.22
Day 7	8.42 ± 0.22	8.43 ± 0.20	8.66 ± 0.23	8.60 ± 0.22
Day 21	14.01 ± 0.28 ^a^	12.11 ± 0.47 ^b^	13.80 ± 0.49 ^a^	12.64 ± 0.41 ^b^
Average daily weight gain (g)				
Day 0 to 7	256.43 ± 7.10	258.22 ± 11.50	290.89 ± 5.35	281.79 ± 14.90
Day 7 to 21	399.64 ± 8.86 ^a^	262.50 ± 20.5 ^b^	366.79 ± 19.22^a^	288.84 ± 22.36 ^b^
Average daily feed intake (g)				
Day 0 to 7	313.42 ± 8.12 ^b^	311.75 ± 7.87 ^b^	331.97 ± 5.23 ^a^	320.45 ± 9.50 ^b^
Day 7 to 21	566.37 ± 12.53 ^a^	462.47 ± 14.90^c^	560.30 ± 11.18 ^a^	500.78 ± 11.70 ^b^
Feed to gain ratio				
Day 0 to 7	1.23 ± 0.04	1.23 ± 0.07	1.14 ± 0.02	1.16 ± 0.08
Day 7 to 21	1.42 ± 0.04 ^b^	1.81 ± 0.10 ^a^	1.55 ± 0.06 ^ab^	1.78 ± 0.11 ^a^

Data are means ± standard error; *n* = 8 for each group. Mean values sharing different superscripts within the same row differ significantly (*p* < 0.05). Control, piglets fed the basal diet without diquat injection; diquat, piglets fed the basal diet and challenged with diquat; 0.15% Trp + diquat, piglets fed the 0.15% supplemented diet and treated with diquat; 0.30% Trp + diquat, piglets fed the 0.30% supplemented diet and treated with diquat.

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
