# Peer review of "L-Tryptophan Enhances Intestinal Integrity in Diquat-Challenged Piglets Associated with Improvement of Redox Status and Mitochondrial Function"

_animals, 2019, doi:10.3390/ani9050266_

Round 1

Reviewer 1 Report

Comments to the Authors of manuscript number: animals-509168 entitled L-tryptophan enhances intestinal integrity in diquat-2 challenged piglets associated with improvement of 3 redox status and mitochondrial function.

In the present paper I reviewed effects of tryptophan on intestinal barrier are presented. As the authors have pointed out, studies on dietary tryptophan are needed due to the lack of described "appropriate" dose.

What is more disturbing is the fact of the use of diquat (or diquat dibromide, is the common name of the chemical 6,7-dihydrodipyrido (1,2-a:2',1'-c) pyrazinediium dibromide). Diquat, thiram and pymetrozine have been banned by the European Commission. However, Authors wrote about diquat in the first sentences of Simple summary and explain why this chemical agent was used. I wondered for a long time what decision to make with regard to further fate of the manuscript. I checked and diquat is commonly used in many different studies conducted on animals. Further, this manuscript should be corrected, but is well written. I recommend minor revision.

1. L48- “functionally regulates host physiology and metabolism”

Authors should describe clearly.

2. L57, L 79, L85....and others – authors wrote “an appropriate dosage”

the use of the word is inappropriate. Anything can be a drug or a poison, and it depends on the appropriate dose.

3. L96 – were animals from the same litter? The choice of the animal should be described in details.

4. L98 – why they were castrated?

5. L100 – Was the body weight of each piglet the same? How were the piglets selected?

6. part of "Animals and treatment".

Authors should explain why there is no the group supplemented with Trp?

7. L112 – “These three diets were kept isonitrogenous and isoenergetic .....” what about control diet?

There should be information that this diet also was isonitrogenous and isoenergetic.

8.  L122 Feed consumption is not the same every day. How Authors have calculated feed intake of piglets were weighed only two times at d 7 and 21? It should be explained.

How single result could be used to calculate ADFI for 7 days?

The same for ADG.

9. L123-124 How piglets were weighted?

10. Table 1. “CP” and “AP” should be explained.

11. Table 1. Which components have been calculated and which have been analyzed.

12. If the authors give p in the text then they must give the exact value.

13. L351- these proteins are different from those of the part of “Gel electrophoresis and Western Blotting”

14. Authors should try to explain why dietary supplementation with 0.15% Trp is more effective than dietary supplementation with 0.3% Trp?

Authors describe the results of other works, they explain much of oxidative stress, the intestinal barrier, but there is no why lower dose is better.

Author Response

Dear reviewer,

Thanks for your efforts on our manuscript. We really appreciated that you can give us the opportunity to improve and revise our paper. All the changes were marked in red in the revised manuscript. The point to point response record was attached. Thanks again.

Reviewer 2 Report

Title: L-tryptophan enhances intestinal integrity in diquat-2 challenged piglets associated with improvement of 3 redox status and mitochondrial function.

Comments:

Results and discussion

-        Explain why with 0.30% tryptophan no effects are observed. The explanation that 0.15% is closest to the recommendation is superficial.

-        Explain the mechanism by which tryptophan improves productive performance.

-        Describe the possible physiological and / or biochemical mechanisms that explain that tryptophan improves Tight junction and antioxidant activity in the intestine.

References

-        There are several errors in the format of the references. Review all references carefully.

-        Ref. 1: change J Anim Sci. by J Anim Sci (without the point).

-        Ref. 2: change 51,215-236 by 51,215-236.

-        Ref. 4: change 90,2264–2275by 90, 2264–2275.

-        Ref. 12: journal without the point.

-        Ref. 16: After the paper title goes a point, not a comma.

-        Ref. 17: After the paper title goes a point, not a comma.

-        Ref. 31: title of the paper only the first letter in capital letters.

Author Response

(The authors gave the same response as above.)
